# Mapping Geospatial Processes Affecting the Environmental Fate of Agricultural Pesticides in Africa

**DOI:** 10.3390/ijerph16193523

**Published:** 2019-09-20

**Authors:** Chantal M. J. Hendriks, Harry S. Gibson, Anna Trett, André Python, Daniel J. Weiss, Anton Vrieling, Michael Coleman, Peter W. Gething, Penny A. Hancock, Catherine L. Moyes

**Affiliations:** 1Big Data Institute, Li Ka Shing Centre for Health Information and Discovery, University of Oxford, Oxford OX3 7LF, UK; harry.gibson@bdi.ox.ac.uk (H.S.G.); andre.python@bdi.ox.ac.uk (A.P.); daniel.weiss@bdi.ox.ac.uk (D.J.W.); peter.gething@bdi.ox.ac.uk (P.W.G.); penny.hancock@bdi.ox.ac.uk (P.A.H.); catherinemoyes@gmail.com (C.L.M.); 2Team Sustainable Soil Use, Wageningen Environmental Research, P.O. Box 47, 6700 AA Wageningen, The Netherlands; 3Department of Vector Biology, Liverpool School of Tropical Medicine, Liverpool L3 5QA, UK; anna.trett@lstmed.ac.uk (A.T.); Michael.Coleman@lstmed.ac.uk (M.C.); 4Faculty of Geo-Information Science and Earth Observation (ITC), University of Twente, P.O. Box 217, 7500 AE Enschede, The Netherlands; a.vrieling@utwente.nl

**Keywords:** artificial compound, crop protection, environmental data, insecticide residue, satellite data, tropics

## Abstract

The application of agricultural pesticides in Africa can have negative effects on human health and the environment. The aim of this study was to identify African environments that are vulnerable to the accumulation of pesticides by mapping geospatial processes affecting pesticide fate. The study modelled processes associated with the environmental fate of agricultural pesticides using publicly available geospatial datasets. Key geospatial processes affecting the environmental fate of agricultural pesticides were selected after a review of pesticide fate models and maps for leaching, surface runoff, sedimentation, soil storage and filtering capacity, and volatilization were created. The potential and limitations of these maps are discussed. We then compiled a database of studies that measured pesticide residues in Africa. The database contains 10,076 observations, but only a limited number of observations remained when a standard dataset for one compound was extracted for validation. Despite the need for more in-situ data on pesticide residues and application, this study provides a first spatial overview of key processes affecting pesticide fate that can be used to identify areas potentially vulnerable to pesticide accumulation.

## 1. Introduction

The environmental fate of agricultural pesticides can have direct and indirect impacts on human health and the environment. For example, human exposure to toxic levels of dichlorodiphenyltrichloroethane (DDT) can result in spontaneous abortion by women [1], carbamate and organophosphate in the environment can result in biodiversity loss [2], and there is evidence that pesticide exposure can play a role in neurodegenerative conditions like dementia [3] and Parkinson’s disease [4]. Agricultural insecticides can also drive the spread of resistance in non-target insects that are involved in the transmission of human diseases such as malaria and dengue [5,6]. In this instance, agricultural pesticides have an indirect impact on human health by reducing the efficacy of insecticide-based interventions for disease control. 

It is known that agricultural pesticides are regularly being used in African farming systems [7,8]. Pesticide application in Africa is projected to increase by 1.2 to 2.3 times due to economic and population growth and due to the few countries that have implemented regulations and legislations on pesticide application [9]. Although the average pesticide use is still relatively low in Africa, the potential negative effects on human health and the environment are substantial [8]. This is mainly due to illiteracy among farmers, lack of awareness about the danger of pesticide misuse, difficulties with extrapolating the prescribed pesticide dose ratio to the size of an agricultural field, and lack of knowledge on pests and diseases [10]. 

To understand where in Africa the risk of pesticide exposure is highest, information is required on (i) where pesticides are sprayed and (ii) where pesticide residuals subsequently accumulate. This study focuses on the second of these factors; where will pesticides accumulate if they are sprayed in the environment? A continental analysis of the areas where the environment is vulnerable to pesticide accumulation is a first step in mapping the potential risk of pesticide application. To identify the areas that are most vulnerable to pesticide accumulation, this study aims to map processes associated with the environmental fate of agricultural pesticides using publicly available geospatial datasets. Mapping this vulnerability by adapting or developing pesticide fate models is difficult for Africa because most models were developed for temperate climates and pesticide behaviour in tropical environments is generally less understood compared to temperate environments [11,12,13]. 

The processes that different pesticide fate models use to assess the environmental fate of agricultural pesticides were reviewed to identify the processes that are important to map. Key processes were then selected based on the study’s aim of mapping the spatial variation in the most important processes across the continent of Africa. Specifically, processes were selected that were commonly used by a range of pesticide fate models, that are relevant at a resolution of 2.5 arc-minute (approximately 5 km) and that apply to pesticides generally (as opposed to pesticide-specific processes such as transformation and degradation). Geospatial datasets were then sourced to create maps showing the spatial variation in each process. 

## 2. Materials and Methods

### 2.1. Selection of Key Processes Identified by Pesticide Fate Models

To identify the processes to map in this study, the processes identified as important by a range of pesticide fate models were reviewed. Pesticide fate models operate at different scales, for different purposes and at different levels of detail. Therefore, the processes included in each model differ as well. We first identified relevant existing pesticide fate models, then we listed the processes that each pesticide fate model uses. From this list, a subset of processes was selected based on their importance, relevance to mapping at a 5 km scale across Africa, and the feasibility of mapping each process using publicly available geospatial datasets.

#### 2.1.1. Identify Pesticide Fate Models

From the large number of available pesticide fate models, we identified those that provided an overview of most relevant processes affecting the environmental fate of pesticides, giving preference to models of relevance to African environments. Different sources were consulted to identify available pesticide fate models and their relevance for Africa. Models that were applied or developed, calibrated and validated in tropical areas were identified using the Web of Knowledge and the search term ‘pesticide fate model’ AND ‘tropic *’. Other pesticide fate models were found by consulting CEAM (Center for Exposure Assessment Modeling), OPPT (Office of Pollution Prevention and Toxics), CEMC (Canadian Environmental Modelling Centre), FOCUS (Forum for the Co-ordination of pesticide fate models and their Use), OECDs (Organization for Economic Co-operation and Development) model database, RIVM (National Institute of Public Health and the Environment) and WEnR (Wageningen Environmental Research). Two review papers were also used [14,15]. 

We only selected models that: (i) operate at catchment scale or coarser, (ii) operate at daily scale or coarser, (iii) were not developed for one specific process or crop. We also discarded complex models that required data at fine resolution, many different input sources (e.g., SWMS_3D (Simulating Water and solute Movement in three-dimensional variably Saturated media), FEHM (Finite Element Head Model)), and models that were derived from a combination of other pesticide fate models. We thus retained a total of 24 models (Table 1).

#### 2.1.2. Selecting Key Processes Affecting Pesticide Fate

The processes used by the 24 pesticide fate models identified in Table 1 were listed (Appendix A). A total of 20 processes were used in one or more of the 24 pesticide fate models. A subset of these 20 processes were selected based on the following criteria: (i) inclusion in at least ten of the selected pesticide fate models, (ii) relevant at the resolution and extent of this study, i.e., a 2.5 arc-minute resolution applied across Africa, (iii) relevant to the fate of pesticides after application (as opposed to factors related to the application rate), and (iv) generally applicable to all pesticides (as opposed to pesticide-specific processes such as transformation and degradation). These criteria resulted in the selection of four key processes: leaching, surface runoff, soil storage and filtering capacity, and volatilization. The criterion of inclusion in at least ten pesticide fate models was relaxed for the process of erosion, which was included in 9 of the 24 models, because erosion may play a more important role in Africa as compared to the countries for which the existing pesticide models were developed. Approximately 25% of the African land surface suffers from water erosion [41]. The combination of high rainfall intensity, sloping land and soils that are, in general, poor in nutrients and organic matter increases erosion risk in Africa [42]. When the flow velocity decreases, soil particles in erosion settle out. Pesticides can bind to soil particles and, therefore, sedimentation was the fifth process selected for this study. 

The key processes selected for this study are visualised in Figure 1 and defined as follows:-Leaching is the process by which rain or irrigation water infiltrates and percolates to deeper groundwater layers.-Surface runoff is the process by which rain or irrigation water flows overland to other streams or surface water.-Sedimentation is the process by which soil particles in suspension settle out of fluid, water in this instance, and come to rest.-Soil storage and filtering capacity indicates the capacity of a soil to store and filter substances (e.g., water or pesticides).-Volatilization is the process whereby a chemical substance is converted from a liquid or solid state to a gaseous or vapour state.

### 2.2. Selection of Geospatial Datsets

Geospatial data were needed to map the spatial variation in the five key processes affecting pesticide fate. Alternative data sources were sometimes available for these data. Climate data can, for example, be obtained from NASA (National Aeronautics and Space Administration) or WorldClim—Global Climate Data, and soil organic matter content can be obtained from the Harmonized World Soil Database (HWSD) or SoilGrids. To select the most suitable data source, priority was given to datasets that: (i) covered Africa and had a resolution of 2.5 arc-minute (approximately 5 × 5km pixels at the equator) or finer, (ii) was most up-to-date, (iii) was established by an agency (e.g., the National Aeronautics and Space Administration) or recognized by other studies, and (iv) was accompanied by a quality assessment. Further details of the selected geospatial datasets are provided in Appendix A. The geospatial datasets were averaged to 2.5 arc-minute resolution before they were used as input data for the models. Some datasets did not cover islands (e.g., Cape Verde, Comoros, Mayotte) or only covered Sub-Saharan Africa, but met the other criteria. In other instances, the dataset used was the only one available. These data limitations meant that the geographical extent of some of the maps created by this study was restricted. 

### 2.3. Mapping Key Processes Affecting Pesticide Fate

The key processes identified in Section 2.1 were mapped using publicly available geospatial data and existing algorithms. Some of these processes are driven by variables for which no geospatial datasets were available. These variables could not be included or were replaced by another, correlated variable for which a geospatial dataset was available. In some instances, an algorithm to calculate values for that process was not available, so the variables known to drive the process were simply combined. Both of these limitations meant that relative spatial variation in each process was mapped, rather than a specific measurement value for that process. 

The next sections explain the approach used for modelling each key process. Table 2 gives an overview of the variables that were required to map spatial variation in the pesticide fate processes, and the variables that were actually used as input data. The key processes were mapped at 2.5 arc-minute resolution, because they were initially constructed for a wider project on insecticide resistance in malaria vectors that operated at 2.5 arc-minute resolution [43].

#### 2.3.1. Leaching

Data on soil drainage rate, groundwater depth, bedrock depth and type, slope, and soil moisture were required to create a map on the geospatial variation in leaching [57,58]. Data on soil drainage class were obtained from AfSoilGrids [44]. The dataset classifies drainage based on soil organic matter content, soil structure, and soil texture. AfSoilGrids combines the Africa Soil Profiles (AfSP) database and the AfSIS (Africa Soil Information Service) Sentinel Site database with explanatory variables to spatially predict soil drainage classes using the random forest method. Low infiltration rates correspond to <15 mm/h, moderate infiltration rates correspond to 15–50 mm/h and high infiltration rates correspond to >50mm/h [59]. 

A global groundwater depth map at 30 arc-second resolution is available [45]. This map is based on limited observations (431 sites) for Africa, but it is the best spatially exhaustive prediction on groundwater depth available. Data on bedrock depth were obtained from SoilGrids [46]. Bedrock type is an indicator for porosity. Leaching takes place more easily in bedrock with high porosity. The porosity of the bedrock is strongly related to the soil drainage rate; therefore, data on the soil drainage class serves as an indicator for bedrock type. Slope was derived from the Shuttle Radar Topography Mission 90m Digital Elevation Database v4.1 [47]. The mean soil moisture content was obtained from National Aeronautics and Space Administration—United States Department of Agriculture (NASA-USDA) Global Soil Moisture Data. These data were only available at 12.5 arc-minute, but because this is the only data on soil moisture available, the selection criterion was relaxed for this geospatial dataset. 

The DRASTIC model used, among others, environmental variables to evaluate the vulnerability of groundwater contamination [60]. Not every variable has the same impact on leaching and, therefore, the DRASTIC model assigns a weight to each variable. The weights that were assigned to the variables in the DRASTIC model have also been assigned to our model, resulting in Equation (1).
(1)L=5D+5(1−GW)+2(1−DB)+(1−SL)+5SM,
where, *L* represents the vulnerability to leaching, *D* is the drainage class, *GW* is the normalised groundwater depth, *DB* is the normalised depth to bedrock, *SL* is the normalised slope and *SM* is the normalised mean soil moisture content between 2010 and 2018. The resulting vulnerability for leaching is adimensional and was normalized between 0 and 1. The DRASTIC model has been applied in multiple studies on groundwater leaching [61,62,63]. Not taking non-linearity into account might result in an over- or underestimation of assessments of geospatial variation in leaching [57,58]. The individual datasets can be combined in more sophisticated ways when knowledge on the relationships between the input data and leaching becomes available for Africa.

#### 2.3.2. Surface Runoff 

Surface runoff was divided into three processes: the susceptibility for surface runoff generation, transfer and accumulation. These processes were modelled based on the Indicator of Intense Pluvial Runoff (IRIP) method. This method creates comprehensive maps of areas susceptible for surface runoff without explicit hydrological modelling [64]. Each process required five variables (Table 2). The method is described in more detail by Lagadec et al. [65]. However, in comparison to this study, we used normalised continuous maps as input data instead of binominal data.

Data on soil drainage rate, soil thickness, soil erodibility, topography, and land use were required to model the spatial variability in surface runoff generation [65]. An existing model described by Wischmeier and Smith [66] was used to obtain a map on the soil erodibility. This method is explained in more detail in Section 2.3.3. The topography indicator of the IRIP method is a combination of slope and topographical wetness index (TWI) and were both derived from the SRTM-DEM (Shuttle Radar Topography Mission-Digital Elevation Model). Land use classes were obtained from the Global Mosaics of the standard MODIS (MODerate resolution Imagine Spectroradiometer) land cover type data product MCD12Q1 [50]. This product collated land use data between 2001 and 2012 and categorized the data into 17 different land use classes. Based on background information [65,67], we categorized the MODIS land cover type data product into five classes and gave a weight to each class to indicate how infiltrative or impervious surfaces under a certain land use class are (Table 3).

Data on surface runoff generation, slope, break of slope, catchment capacity and artificial linear axes were required to model surface runoff transfer [65]. Data on slope were obtained from the SRTM-DEM. Catchment capacity is estimated using the Horton Form factor [68]. This factor is the ratio of area to length of the sub-watershed defined by the drained area at the considered pixel. The area of the watershed and the stream length were both obtained from HydroSHEDS (Hydrological data and maps based on SHuttle Elevation Derivatives at multiple Scale) [69]. The continental extent of our study did not allow for the inclusion of ‘Break of slope’ and ‘Artificial linear axes’. 

Data on surface runoff generation, slope, break of slope, TWI, and flow accumulation were required to model surface runoff accumulation [65]. How the first three indicators were obtained is described above. Flow accumulation was obtained from HydroSHEDS [69]. 

#### 2.3.3. Sedimentation

Data on the erosion rate within a catchment area were required to map geospatial variation in sedimentation. The erosion rate was quantified using the highly acknowledged Universal Soil Loss Equation (USLE) (Equation (2)) [70].
(2)E =R×K×C×LS×P,
where, *E* is the annual average soil loss through water erosion (in t/ha/year), *R* is the rainfall erosivity (in MJ·mm/ha/h/year) that represents the power of rainfall to cause soil erosion by water, *K* is the soil erodibility factor in (t ha h)/(ha MJ mm) that represents the non-resistance of soils to erosion, *C* is the cover-management factor that represents the influence of land use and management on soil erosion, *LS* is the topographic factor that represents the effect of slope-length and steepness on erosion, and *P* is the support practices factor which represents the effects of human practices on erosion prevention. The USLE equation was chosen because it requires relatively little input data and most of these input data can be obtained from geospatial datasets. 

The global rainfall erosivity map [51] was used to represent the rainfall erosivity factor. In this study, a Global Rainfall Erosivity Database was compiled and Gaussian Process Regression was applied to construct the rainfall erosivity map. The soil erodibility factor was estimated by Equation (3) [66].
(3)$$K=\left[\frac{2.1\times 10^{-4}M^{1.14\;}\left(12-\mathrm{OM}\right)+3.25\left(s-2\right)+2.5\left(p-3\right)}{100}\right]\times \;0.1317,$$
where, *M* is the textural factor calculated by Equation (4), *OM* (%) is the organic matter content, *s* is the soil structure class where 1 is very fine granular, 2 is fine granular, 3 is medium or coarse granular and 4 is block, platy or massive, and *p* is the soil drainage class.
(4)$$M=m_{silt}+m_{vfs}\times \left(100-m_{c}\right),$$

In Equation (4), m_silt_ (%) is the silt fraction (0.002–0.005 mm), m_vfs_ (%) is the very fine sand fraction (0.05–0.1 mm), which equals 20% of the sand fraction, and m_c_ is the clay fraction (<0.0002 mm). Data on soil texture, organic matter content and drainage class were obtained from SoilGrids [46]. Data on soil structure were obtained from the HWSD [52]. 

The slope-length factor (LS) depends on two components: slope and length of the slope. This study only considered the component slope, because the length of the slope affects erosion rate at much finer resolution [71] than the 2.5 arc-minute that was used in our study. Including the length of the slope would increase the error. To estimate the slope-factor (*S*), distinction was made between slopes flatter than 9% (Equation (5a)) and steeper than 9% (Equation (5b)) [72].
(5a)S=10.8×sin(θ)+0.03  if slope<9%,
(5b)S=16.8×sin(θ)−0.5  if slope≥9%,
where θ is the slope in degrees. 

The cover-management factor required data on land management, which was not available for the African continent. Therefore, the enhanced vegetation index (EVI) was assumed to be a good proxy for the cover-management factor [73]. Gap-filled mean EVI data were available for Africa [53]. This study extracted data for the African continent from the MODIS Enhanced Vegetation Index (EVI) dataset, and daytime and night-time Land Surface Temperature (LST) datasets, and applied two complementary gap-filling algorithms and a variety of run-time options to create data on the EVI. No spatial data on support practices were available for Africa and, therefore, the factor was excluded in the model.

Applying the USLE equation gave an estimation of the erosion rate across Africa. The sediment load within a watershed is calculated by the sum of erosion within a watershed. We assumed that sediment transport and water transport are steered by the same factors. Sedimentation takes place in the streams where flow velocity is low. The flow velocity of water can be calculated by combining the sediment load per watershed and slope Equation (6), [74].
(6)$$V=\;V_{m}\frac{s^{b}A^{c}}{{\left[s^{b}A^{c}\right]}_{m}},$$
where V is the stream velocity, V_m_ is the average stream velocity within a watershed which is assumed to be related to the mean slope within a watershed, s is the slope, A is the flow accumulation, b and c are constants which are both fixed at 0.5 in this study and [s^b^A^c^]_m_ is the average slope-area term within a watershed. The map on flow velocity can now be combined with the map on sediment load to identify most vulnerable areas for sedimentation. 

#### 2.3.4. Soil Storage and Filtering Capacity

The soil storage and filtering capacity is influenced by the soil organic matter content, clay content, soil pH and cation exchange capacity (CEC) [75]. The spatial patterns of filtering capacity were similar to those of storage capacity according to Makó et al. [75] and, therefore, one map was constructed for both. 

All input data were obtained from SoilGrids [46]. This data source provided soil characteristics at seven fixed-depth intervals ranging between 0 to 200 cm. Soil profile data were obtained by taking depth-weighted averages of these seven layers. Although we know which environmental data were influencing the storage and filtering capacity, the relationships between these data and storage and filtering capacity are location- and pesticide-specific [75]. Lack of data on storage and filtering capacity in African soils also hampers the use of statistical algorithms to find the best relationship. Therefore, the data were combined using a linear relationship (Equation (7)).
(7)SFC=OC+C+(1−pH)+CEC,
where *SFC* is the soil storage and filtering capacity, *OC* is the normalised organic carbon content and *C* is the normalised clay content. Soil *pH* and *CEC* were also normalised. Areas where the *SFC* is low are more susceptible to pesticide fate. In reality, the relationship between the variables and the soil storage and filtering capacity is complex, non-linear and depends strongly on the chemical and physical composition of the pesticide [76]. The variables can be combined in more sophisticated ways when data on the applied pesticide compound and knowledge on the relationships between the variables and the soil storage and filtering capacity become available.

#### 2.3.5. Volatilization

Data on potential evapotranspiration (PET), wind speed, air temperature, solar radiation and relative humidity were required to map volatilization [77]. Long-term annual average PET data were obtained from the CSI-CGIAR Global Potential Evapotranspiration Climate Database [54]. Long-term (1970–2000) average monthly wind speed and solar radiation data were obtained from WorldClim V.2 [55]. Monthly maps on the average land surface temperature were derived from daily data MODIS product MOD11A1 V6. Data on relative humidity between 2015 and 2018 were obtained from the Global Forecast System (GFS) of the National Centers for Environmental Prediction (NCEP). Based on these years, average monthly relative humidity was estimated. Again, due to a lack of knowledge on the relationship between these data and the volatilization rate, the key variable associated with volatilization was estimated using Equation (8).
(8)$$V_{i}=WV_{i}+S_{rad,i}+{\;T}_{i}\;+\;\mathrm{PET}+\left(1–{\;\mathrm{RH}}_{i}\right),$$
where, *V_i_* is the key variable associated with volatilization in month *i*, *WV_i_* is normalised long-term wind velocity in month *i*, *S_rad,i_* is the normalised long-term solar radiation in month *i*, T*_i_* is the normalised long-term average day-time surface temperature in month *i*, PET is the normalised long-term annual average potential evapotranspiration and RH*_i_* is the normalised average relative humidity in month *i*. Volatilization is strongly influenced by the dose and method of pesticide application [78] and we know that the relationship between the variables and volatilization are non-linear [78]. We recommend combining the variables in more sophisticated ways when data and knowledge on the relationships between the input data and volatilization becomes available for Africa.

### 2.4. Testing the Maps Associated with Pesticide Fate

Ideally, each map should be validated using observational data for the process that was mapped. However, the data for any of the key processes were not readily available. Therefore, the maps created in this study could not be validated by standard methods. In addition, (i) an insecticide residue database was compiled to test whether the maps could be used as variables in an insecticide residue prediction model, and (ii) a sensitivity analysis on the variables and parameters of the model was carried out. 

#### 2.4.1. Insecticide Residue Database

This study was part of a wider project on insecticide resistance [43] and, therefore, an observational dataset on insecticide residues was compiled for Africa. The dataset was compiled from a literature review in Web of Knowledge to identify studies that measured insecticide residues in soil, sediment, water and air. The search terms that were used and the resulting database are available in Appendix A. The following data were systematically extracted from individual papers: year and month(s) of sampling, sample collection methods and depth, insecticide extraction method, insecticide quantification method, quantification and detection limits, insecticide and insecticide class, the measured insecticide concentration and geographical coordinates of the residue collection site. 

The database contained 10,076 observations, of which 9867 could be georeferenced. The observations were collected from 68 studies. Within this database, 93 different types of insecticides were measured in 2344 soil samples, 3163 sediment samples, 3874 water samples and 486 air samples. A lack of standardisation in the collection, extraction and detection methods made it difficult to construct a standard dataset for use in testing the maps constructed here. The number of samples that were measured at unique locations dropped rapidly when a single insecticide was selected. Figure 2 provides an example for the insecticide compound that was most frequently measured in soil, sediment, water and air. 

#### 2.4.2. Using the Created Maps to Spatially Predict Insecticide Residues

Overall, pp’Dichlorodiphenyldichloroethane (pp’DDD) was most frequently and most consistently measured in two substrates: soil and sediment. Therefore, pp’DDD observations measured in soil and sediment were extracted from the database to obtain a standard dataset for one compound. This resulted in the extraction of 169 and 216 observations measured at 100 locations between 1992 and 2016 across the whole continent. The limited number of observations and their clustered locations made it impossible to perform a robust machine learning modelling analysis of the ability of the maps constructed here to predict spatial variation in insecticide residues. 

#### 2.4.3. Sensitivity Analysis on Variables and Parameters

Some processes were modelled using simple linear relationships because often the relationship depends on the chemical and physical composition of the pesticide or there was no evidence of non-linearity. The output of the models is related to the uncertainty in the variables and the parameters that were given to the model. To test the sensitivity of the variables and the parameters, we carried out a one-at-a-time sensitivity analysis. We changed each variable one by one by 5% and we calculated the average change and the variation in change. The parameters of the process associated with leaching differed between 1 and 5. The effect of changing the parameters on the resulting map were analysed.

## 3. Results

### 3.1. Identifying Pesticide Fate Models and Select Key Processes

Only three out of 24 identified models were developed, calibrated and validated in tropical or sub-tropical areas: the Dynamic Multimedia Environmental Fate Model [22] was developed for the tropical floodplains of Brazil, the Chemical Fate Model [18] was developed for a tropical river catchment in Australia and the Pesticides RIsks in the tropics to Man, Environment and Trade Pesticide model (PRIMET) [34] was developed in Southeast Asia and later adapted to Ethiopia (PRIMET-Ethiopia) [79]. Some models were developed elsewhere, but applied in tropical and sub-tropical areas. For example, the Soil and Water Assessment Tool (SWAT) model [80,81] was developed in the U.S.A., but had, for example, frequently been applied in Southeast Asia. The Pesticide Root Zone Model (PRZM) [82] and the TOXic substances in Surface Waters (TOXSWA) model [35] were developed in the U.S.A and The Netherlands, respectively, but the models have been applied in Ethiopia [83]. The Environmental/Policy Integrated Climate (EPIC) model [23] was developed in the USA, but has, amongst others, been applied in West Africa and Brazil [84], and the Coastal Zone Model for Persistent Organic Pollutants—Version 2 (CoZMo-POP-2) model [20] was also developed in the U.S.A., but has been applied in Botswana [85]. Most of the identified pesticide fate models were not developed in or for Africa. As a consequence, we had to assume that the selected key processes affecting pesticide fate were also key for Africa. 

### 3.2. Mapping Key Variables Associated with Pesticide Fate

#### 3.2.1. Leaching

The spatial variation estimates of leaching resulting from Equation (1) is normalised in Figure 3. The estimates are highest in Central Africa and in the southern coast of West Africa (Figure 3). The tropical climate of these regions causes high soil moisture contents throughout the year, which has a positive effect on leaching. The regions are also characterized by relatively shallow slopes and low elevation. Steeper and higher areas with arid or semi-arid climate are less prone to leaching, e.g., the Great Rift Valley. 

The model does not correct for the more rapid infiltration caused by cracked clay soils. It is known that the hydraulic processes of these soils differ from any other soil [86]. These soils, i.e., Vertisols, are especially common in East Africa. The effect of leaching may, therefore, differ in this part of Africa.

#### 3.2.2. Surface Runoff

According to our results, surface runoff generation was highest in areas where soil permeability was low and bedrock was near the surface (Figure 4A). Steep slopes and high susceptibility for surface runoff generation made Ethiopia especially vulnerable for surface runoff transportation (Figure 4B) and accumulation (Figure 4C). Many studies have confirmed high rates of surface runoff in Ethiopia [87,88]. 

#### 3.2.3. Sedimentation

The areas that are estimated as most prone to erosion and sedimentation processes are in Ethiopia, the southern and eastern parts of the Democratic Republic of the Congo and Madagascar (Figure 5). In some of these areas, we estimate up to 45 t/ha/year soil erosion. Previous studies confirm that these processes take place in large amounts in these areas. For example, the soils of Madagascar tend to be vulnerable to erosion [89], the Upper Blue Nile Basin (Ethiopia) receives large quantities of sediments from agricultural areas in the catchments [90,91] and natural processes dominate the soil allocation in Congo [92], although agricultural development and deforestation has increased the sediment load over recent decades [93]. The maps for the process associated with erosion was compared to the global erosion map of the European Soil Data Centre (ESDAC). The normalised maps showed a Spearman correlation coefficient of 0.16 and a RMSD of 0.06. The low correlation coefficient must have been caused by differences in input data (e.g., source, resolution), because both methodologies were based on the USLE equation.

#### 3.2.4. Soil Storage and Filtering Capacity

Soil storage and filtering capacity is estimated to be moderate to high in Central Africa, the southern part of West Africa and the Ethiopian Highlands (Figure 6). These regions have relatively high organic carbon (OC) content, clay content and CEC and a low soil pH. The Ethiopian Rift Valley and the Sahara, Namib and Kalahari Desert have lowest storage and filtering capacity. In general, the soils of these areas have extremely low OC contents, are coarser in texture and have a higher soil pH. Pesticide leaching is a minor problem in deserted regions, because of the limited agricultural activity. However, the resilience of soils with a low binding capacity is low, which can affect its bio-functioning [94]. 

The role that soil characteristics play in pesticide binding is less documented and, in general, less understood for tropical soils [95,96,97]. Soil storage and binding capacity depends strongly on the chemical composition and the half-life of the pesticide. Pesticides can have a positive or negative charge or they can be non-polar. Differences in the chemical structure of individual pesticides were beyond the scope of the current study.

#### 3.2.5. Volatilization

The map on mean estimates of the process associated to volatilization vulnerability showed highest values in the Rift Valley, the Horn of Africa and the Namib and Kalahari Desert, and lowest values in the tropical regions and in the Central Highlands (Figure 7A). The standard deviation of the monthly values during the year was highest in areas with inter-annual variation in temperature and relative humidity, and lowest in the Rift Valley and Central Africa (Figure 7B). 

One of the factors that influence volatilization is wind velocity. We used the mean annual wind velocity in the model, although farmers will attempt to reduce spray drift and volatilization by spraying on days when the wind velocity is low. There is also no consistency in the duration and extent of volatilization, because it depends, amongst others, on the application method and environmental conditions. Some studies measured pesticide concentrations only up to a few meters from the source [98] and only for a few hours after spraying [99], while other studies measured pesticides up to a few kilometers from the source [99] and up to two months after spraying [100]. These examples indicate that, in some cases, monthly maps at 2.5 arc-minute resolution might be too coarse for studying the effect of volatilization on pesticide fate. Volatilization is strongly influenced by the dose and method of pesticide application [78]. 

#### 3.2.6. Sensitivity Analysis on Variables and Parameters

The results of the sensitivity analysis are provided in Table 4. The process associated with leaching is most sensitive to groundwater depth (6.2%) and soil moisture (5.8%). Soil thickness (2.1%), surface runoff generation (3.7%) and flow accumulation (2.1%) are the most sensitive variable in the process associated with surface runoff generation, transfer and accumulation, respectively. The variables associated with sedimentation showed very low sensitivity. The soil pH is most sensitive in the process associated with the storage and filtering capacity. The variables solar radiation, temperature, and potential evapotranspiration are almost equally sensitive. Changing the parameters of the model associated with leaching by one point, resulted in a change of 12%. Deviation 2, 3 or even 4 points from the given parameters resulted in a change of 24.1% (4.2%), 36.1% (6.4%) and 48.1% (8.5%), respectively. 

## 4. Potential and Limitations of the Created Maps and Future Perspective

This study has produced maps that show spatial variation in five major processes associated with pesticide fate. Specifically leaching, surface runoff (generation, transportation and accumulation), sedimentation, soil storage and filtering capacity, and volatilization. Variation can be seen at the 2.5 arc-minute scale, for example, in sedimentation across the continent, and at larger scales, for example, higher values for leaching are apparent across the Congo Basin. The spatial variation provided in these maps can play an important role in determining where pesticides end up in areas that have been sprayed in the past or that may be sprayed in the future. 

The major limitation of these maps is that they have not been validated against measurements for these five processes taken from field sites across Africa, because such data are not available. Published methods were used where available. Some of these methods were validated, but for most methods, a sensitivity analysis of the variables was carried out. For example, the weights of the DRASTIC model were used to create the map associated with the process leaching. The DRASTIC model is typically evaluated by a map removal and single-parameter sensitivity analysis [101]. Ouedraogo et al. [102] applied the DRASTIC model at pan Africa scale to analyse groundwater contamination by nitrate. The study concluded that: (i) further validation using more observations is needed, (ii) a better understanding of the factors influencing nitrate contamination is needed and (iii) data monitoring experiments are needed for calibration and validation of the model. The IRIP method that was used to create the map associated with surface runoff was compared to four different datasets: two regulatory zonings of surface runoff and erosion risks and two datasets of surface runoff impacts on roads and railway networks [65]. The correlation coefficients between the four datasets and the surface runoff susceptibility map ranged between 0.64 and 0.8 [65]. The USLE equation is a widely acknowledged and applied equation [103]. The performance of the model depends on its application. For example, considering rainfall erosivity and vegetation cover at shorter time steps can improve the modelled estimates [104]. Auerswald et al. [105] states that validation of USLE-based erosion modelling at regional or larger scales is hampered by the lack of long-term field-scale measurements. Alternatively, a cross-comparison of the model estimates was done using Markov Chain Monte Carlo (MCMC) simulations. This resulted in an error of the of 8 PG/year for the map that estimated global erosion between 35 and 35.9 Pg/year [103]. For creating the map associated with the soil storage and filtering capacity, information was obtained from European maps on soil storage and filtering capacity. However, these maps were not validated [75]. When more knowledge on the structure of the relationship between the variables and the mapped process becomes available, a sensitivity analysis on the complexity of the model can be carried out. 

The models that were used to create the maps associated with pesticide fate used several assumptions. The model associated with leaching assumes, for example, that the weights that were defined by Pérez-Lucas et al. [60] could be applied to Africa. The exclusion of the break of slope from the model associated with surface runoff transfer and the support practices factor from the model associated with sedimentation could have affected the quality of the maps. As soon as these data become available for Africa, we recommend incorporating these variables into the models. Although studies showed a correlation between EVI and land management [106], using EVI as a proxy for land management could have influenced the map associated with sedimentation. 

The processes mapped in this study are all key in determining pesticide fate after an area is sprayed, however, further information is needed before a complete picture of pesticide fate can be constructed or the maps can be combined. Data associated with soil storage and filtering capacity and volatilization were combined in a simplistic way despite the complex non-linear relationships involved because insufficient information was available to model these relationships. Factors linked to the structure of individual pesticides, such as degradation, were not mapped. In addition, other variables such as microbial composition and plant uptake go far beyond the scope of the data available. Long-term monthly averages were not always available from existing geospatial datasets. Therefore, the maps created did not account for the seasonal effect of pesticide fate processes, while it is known that seasonality plays a role in some of the processes [101,104]. Furthermore, creating each pesticide fate process individually does not account for interactions between different processes, which is taken into account by pesticide fate models. 

Finally, in order to map pesticide residues in African environments, information about pesticide fate processes needs to be combined with information on pesticide application. However, data on pesticide application in Africa are sparse and inconsistently collected through space and time. For example, registered governmental data and the Living Standards Measurement Study–Integrated Surveys on Agriculture (LSMS–ISA) database contains data on pesticide application that were outdated, not covering the whole continent, and typically underestimating the actual pesticide use [107,108]. The use of pesticides in Africa should be monitored systematically, e.g., by registering pesticide application, including date, time, type and quantity, to enhance sustainable development. Only three out of 24 pesticide fate models were developed, calibrated and validated for tropical environments. The other models use data from temperate regions [12]. Further investigation is needed to test whether these models can be used for African environments without taking a large number of observations for calibration and validation of the model. The environmental fate of pesticides has locally been studied in Africa. For example, pesticide use in South Africa was mapped [100], surface water contamination in Ethiopia was assessed [83] and the effect of pesticide leaching on the contamination of Lake Naivasha was mapped [102]. These studies provide data that can be used to develop or adapt pesticide fate models for African environments. 

## 5. Conclusions

This study illustrated how geospatial datasets can help in identifying areas vulnerable to pesticide fate when data and knowledge on pesticide application and residues are limited. A set of aquatic, terrestrial and atmospheric processes associated with pesticide fate is provided, which show the spatial variation in the vulnerability across Africa to pesticide accumulation. The study serves as a first step towards pesticide risk assessments for Africa. We recommend using the maps in combination with data on pesticide application or, when it is known which pesticides a crop receives, data on agricultural land use. Systematic data collection for the validation and calibration of pesticide fate models and on pesticide application and residues is essential for future pesticide risk assessments. 

## Figures and Tables

**Figure 1 ijerph-16-03523-f001:**
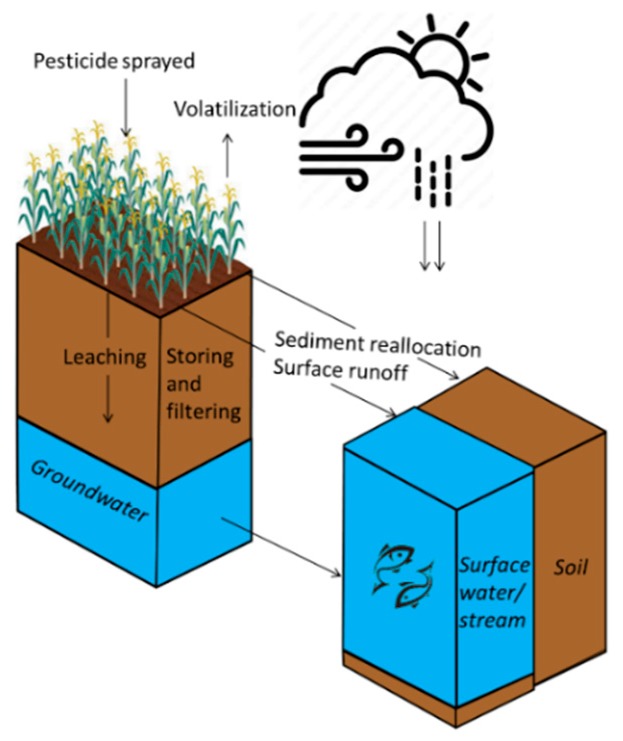
The selected processes affecting pesticide fate and how they act in the environment.

**Figure 2 ijerph-16-03523-f002:**
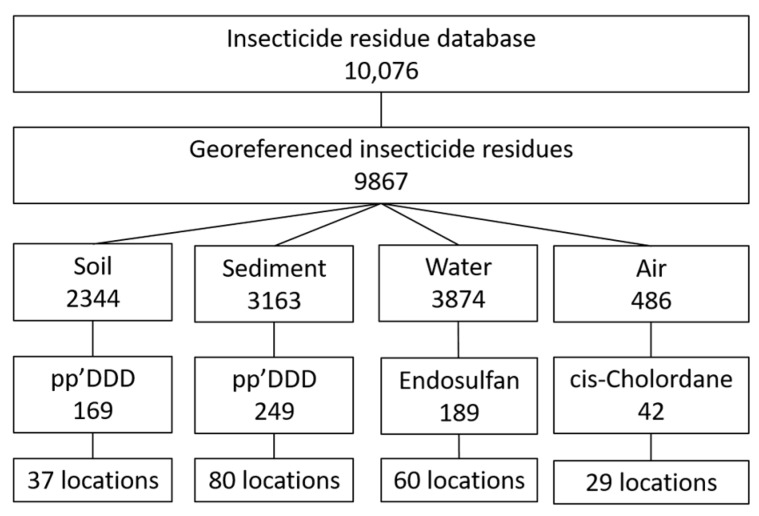
Extracting the number of locations and observations for the insecticide compound that was most frequently measured in soil, sediment, water and air. pp’DDD stands for pp’Dichlorodiphenyldichloroethane.

**Figure 3 ijerph-16-03523-f003:**
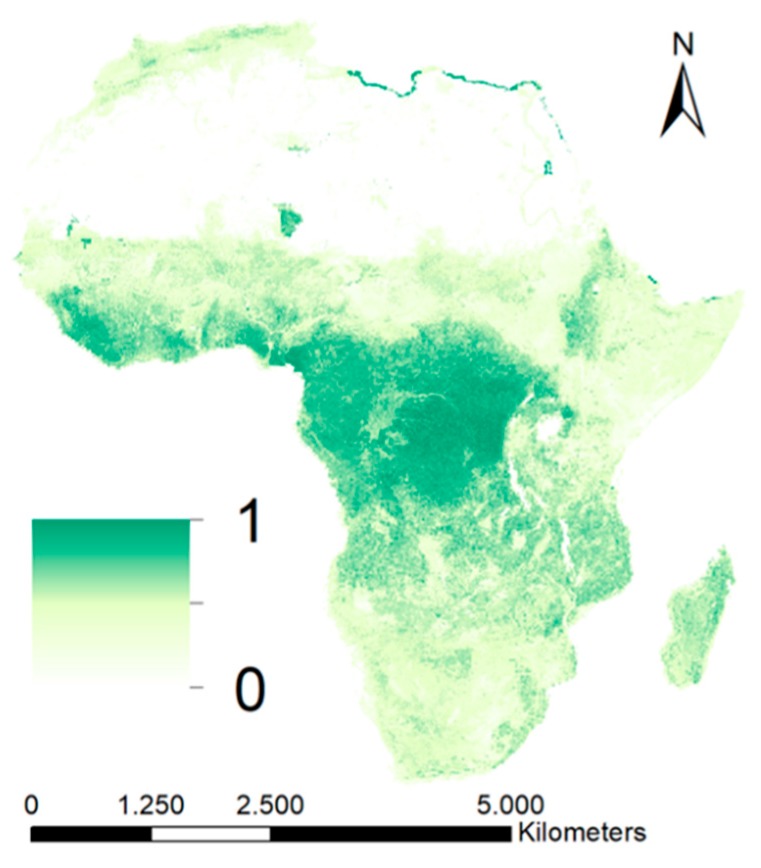
Geospatial variation of the process associated with leaching resulting from Equation (1).

**Figure 4 ijerph-16-03523-f004:**
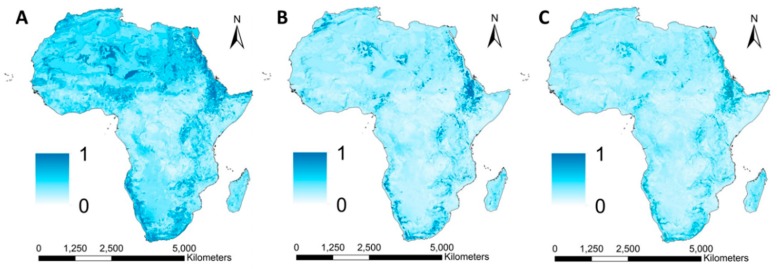
Geospatial variation of the processes associated with surface runoff generation (**A**), transportation (**B**) and accumulation (**C**) assessed by the Intense Pluvial Runoff (IRIP) method (65).

**Figure 5 ijerph-16-03523-f005:**
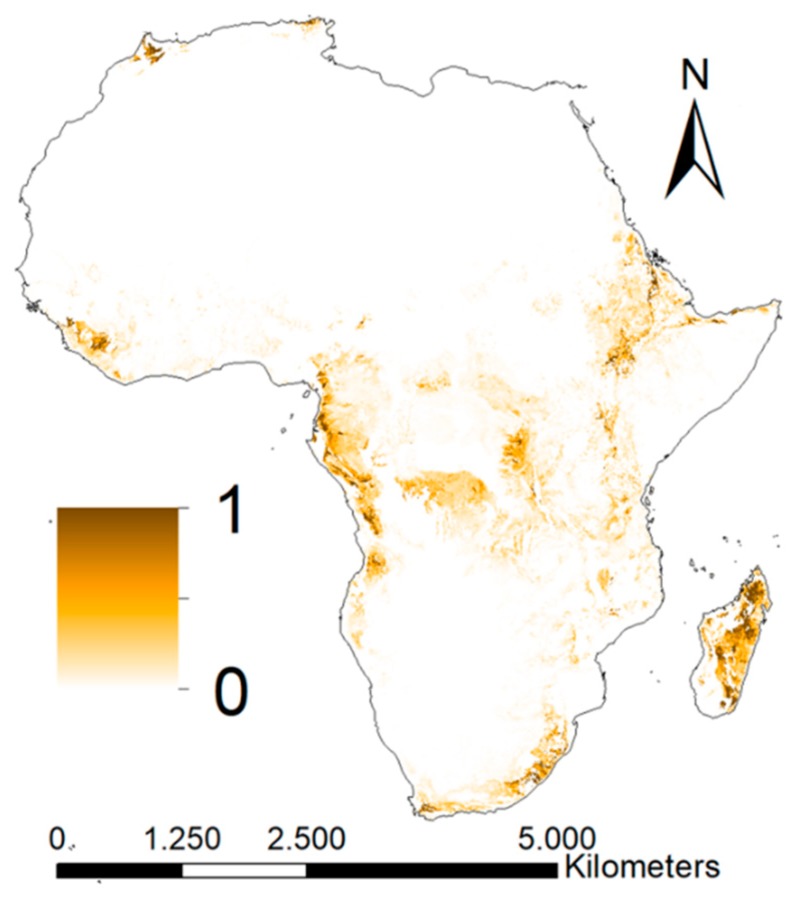
Geospatial variation of the process associated with sedimentation. The map resulted from the Universal Soil Loss Equation (USLE) equation (Equation (2)), the watershed area and the flow velocity (Equation (6)).

**Figure 6 ijerph-16-03523-f006:**
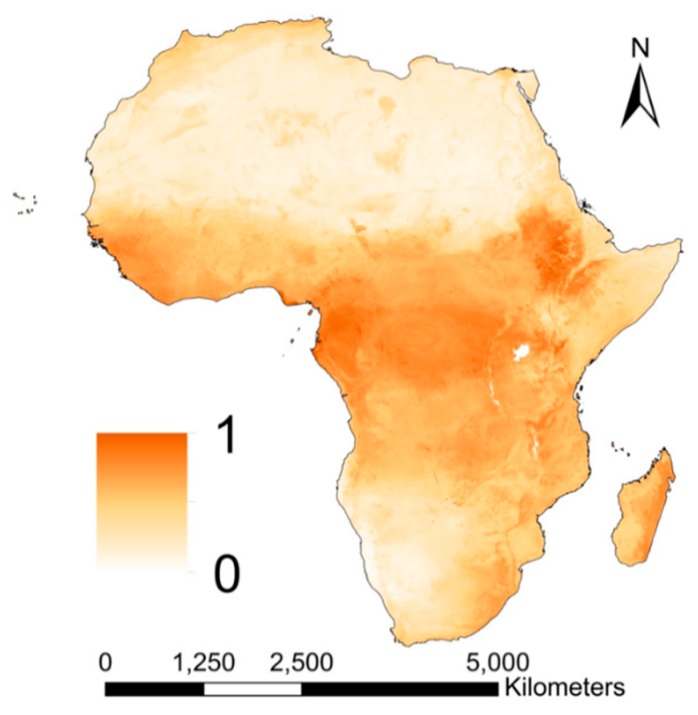
Geospatial variation of the process associated with soil storage and filtering resulting from Equation (7).

**Figure 7 ijerph-16-03523-f007:**
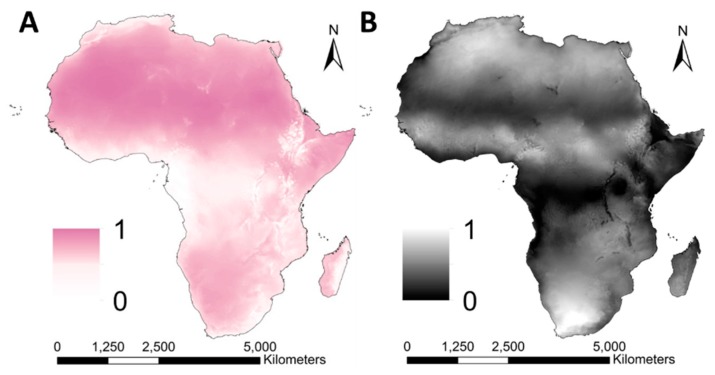
Geospatial variation of the process associated with volatilization; the annual mean (**A**) and standard deviation of the monthly values during the year (**B**). The maps resulted from Equation (8).

**Table 1 ijerph-16-03523-t001:** The pesticide fate models that are identified for this study.

Number	Model Name	Country	Source
1	BASINS	USA	[16]
2	CASCADE-TOXSWA	The Netherlands	[17]
3	Chemical fate model	Australia	[18]
4	CliMoChem	Global	[19]
5	CoZMo-POP-2	USA	[20]
6	CRACK-NP	United Kingdom	[21]
7	Dynamic multimedia environmental fate model	Brazil	[22]
8	EPIC	USA	[23]
9	GIBSI	Canada	[24]
10	GLEAMS	USA	[25]
11	HSCTM-2D	USA	[26]
12	LEACHM	USA	[27]
13	MACRO	Sweden	[28]
14	OPUS	USA	[29]
15	PEARL	The Netherlands	[30]
16	PELMO	Germany	[31]
17	PESTLA	The Netherlands	[32]
18	PLM	United Kingdom	[33]
19	PRIMET	Southeast Asia	[34]
20	PRZM	USA	[35,36]
21	RZWQM	USA	[37]
22	SESOIL	USA	[38]
23	SIMULAT	Germany	[39]
24	SWAT	USA	[40]

**Table 2 ijerph-16-03523-t002:** The geospatial datasets that drive each key process, according to the scientific literature. The geospatial dataset selected by this study for each variable is listed together with the source details. “-“ indicates that no geospatial dataset was available for that variable. “NA” indicates that no source details are given because the dataset was generated by this study.

Pesticide Fate Process	Required Variables	Selected Geospatial Dataset	Source of Geospatial Dataset
Leaching	Soil drainage rate	Soil drainage class	[44]
	Groundwater depth	Groundwater depth	[45]
	Depth to bedrock	Depth to bedrock	[46]
	Type of bedrock	Soil drainage class	[46]
	Slope	Slope	[47]
	Soil moisture	Soil moisture	[48]
Surface runoff—Generation	Soil drainage rate	Soil drainage class	[46]
	Soil thickness	Soil thickness	[49]
	Soil erodibility	Soil erodibility factor	NA
	Topography	Slope	[47]
		Flow accumulation	[47]
	Land use	Land use class	[50]
Surface runoff—Transfer	Surface runoff—Generation	Surface runoff—Generation	NA
	Slope	Slope	[47]
	Break of slope	--	--
	Catchment capacity	Watershed area	[47]
		Stream length	[47]
	Artificial linear axes	--	--
Surface runoff—Accumulation	Surface runoff—Generation	Surface runoff—Generation	NA
	Slope	Slope	[47]
	Break of slope	--	--
	Topographic index	Elevation	[47]
	Flow accumulation	Flow accumulation	[47]
Sedimentation	Rainfall erosivity factor	Rainfall erosivity	[51]
	Soil erodibility factor	Silt content	[46]
		Sand content	[46]
		Clay content	[46]
		Soil organic matter content	[46]
		Soil structure class	[52]
	Cover-management factor	Enhanced Vegetation Index	[53]
	Slope length and slope steepness factor	Slope	[47]
	Support practice factor	--	--
	Erosion	Erosion	NA
	Surface runoff—Accumulation	Surface runoff—Accumulation	
	Watershed area	Watershed area	[47]
Soil storage and filtering capacity	Soil organic matter content	Soil organic matter content	[46]
	Clay content	Clay content	[46]
	Soil pH	Soil pH in H_2_O	[46]
	Cation Exchange Capacity	Cation Exchange Capacity	[47]
Volatilization	Evapotranspiration	Potential evapotranspiration	[54]
	Wind velocity	Wind velocity	[55]
	Temperature	Land surface temperature	[56]
	Relative humidity	Relative humidity	[56]
	Solar radiation	Solar radiation	[55]

**Table 3 ijerph-16-03523-t003:** The weights that were allocated to the different land use classes in order to estimate the process affecting surface runoff.

Forest	0
Grass/scrub/woodland	0.2
Barren/very sparsely vegetated land	0.6
Irrigated and rain-fed cultivated land	0.8
Built-up land	1

**Table 4 ijerph-16-03523-t004:** Each process associated with pesticide fate includes a set of variables. The values represent the average change (in %) and the variation in change (in brackets) of the one-at-a-time sensitivity analysis on the variables, changing them by 5% of the original value.

Process	Variables	−5%	+5%
Leaching	Drainage class	2.4 (0.4)	2.8 (0.5)
	Groundwater depth	6.2 (1.1)	3.2 (0.6)
	Depth to bedrock	2.4 (0.4)	4.1 (0.7)
	Slope	1.2 (0.2)	1.8 (0.3)
	Soil moisture	5.8 (1.0)	2.4 (0.4)
Surface runoff—generation	Soil drainage	1.2 (0.4)	1.2 (0.4)
	Soil thickness	2.1 (0.4)	2.1 (0.4)
	Erodibility	0.3 (0.3)	0.3 (0.3)
	Topography	0.3 (0.1)	0.3 (0.1)
	Land use	1.1 (0.5)	1.1 (0.5)
Surface runoff—transfer	Surface runoff—generation	3.7 (0.9)	3.7 (0.9)
	Slope	1.0 (0.7)	1.0 (0.7)
	Catchment capacity	0.3 (0.7)	0.3 (0.7)
Surface runoff—accumulation	Surface runoff—generation	1.7 (1.3)	1.7 (1.3)
	Slope	0.6 (0.6)	0.6 (0.6)
	Elevation	0.6 (0.6)	0.6 (0.6)
	Flow accumulation	2.1 (2.1)	2.1 (2.1)
Sedimentation	Rainfall erosivity	0.3 (1.0)	0.2 (0.6)
	Soil erodibility	0.6 (1.9)	0.7 (2.4)
	Cropping factor	0.3 (0.9)	0.4 (1.3)
	Slope	0.0 (0.0)	0.1 (0.4)
	Flow velocity	0.0 (0.0)	0.1 (0.3)
Soil storage and filtering capacity	Organic carbon	0.3 (0.4)	0.3 (0.4)
	Clay content	1.2 (0.6)	1.2 (0.6)
	soil pH	4.3 (3.6)	4.3 (3.6)
	Cation Exchange Capacity	0.7 (0.4)	0.7 (0.4)
Volatilization	Wind speed	0.5 (0.2)	0.5 (0.2)
	Solar radiation	1.0 (0.1)	1.0 (0.1)
	Temperature	1.2 (0.2)	1.2 (0.2)
	Potential Evapotranspiration	1.3 (0.2)	1.3 (0.2)
	Relative Humidity	0.7 (0.6)	0.7 (0.6)

## Data Availability

The datasets generated during and/or analysed during the current study are available in the FigShare repositories, Geospatial layers on processes affecting the environmental fate of agricultural pesticides in Africa [109]; Insecticide residue database for Africa [110].

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
