# Peer review of "Mapping Geospatial Processes Affecting the Environmental Fate of Agricultural Pesticides in Africa"

_ijerph, 2019, doi:10.3390/ijerph16193523_

Round 1
Reviewer 1 Report
It would be useful if the captions of the maps would specify the parameters that are mapped with reference to the equations used to evaluate them.
L109 "increase" -> "increases"
L169 "rate and therefore, data" -> "rate, therefore data"
L178 Eq. 1: the "vulnerability to leaching" L is adimentional and varies between 0 and 1? Its values are mapped in Figure 3?
L183 "over- or underestimation of estimates" -> "over- or underestimation of assessments"
L235 "In Eq.3": eq. 4?
L257 Eq. 6: the notation is quite confusing. With reference to Eq. 15 in [44] (page 838) the denominator should be [s^b A^c]_m
L269 "at seven fixed depth intervals ranging between 0 to 200cm depth." -> "at seven fixed depth intervals ranging between 0 to 200cm."
L276 "Areas where the SFC was low are more susceptible to pesticide fate." -> "Areas where the SFC is low are more susceptible to pesticide fate."
L329 Figure 2, caption "pp’DDD stands...": it's not in the figure?
L374 Figure 3: are these values of L of eq. 1?
L397 I do not understand the sentence "Unless both maps were created using nearly the same methodology, differences could have been cause by the input data and the different resolution of the input data.": if the methodology is the same the different output can only be explained by a difference of input (or, to some extent, of resolution).
L461 "Some of these methods were validated, but for most methods a sensitivity analysis of the variables." has been carried out?
Author Response
Response to reviewer 1
It would be useful if the captions of the maps would specify the parameters that are mapped with reference to the equations used to evaluate them.
The captions now refer to the parameters and equations.
L109 "increase" -> "increases"
This has been corrected.
L169 "rate and therefore, data" -> "rate, therefore data"
This has been changed.
L178 Eq. 1: the "vulnerability to leaching" L is adimentional and varies between 0 and 1? Its values are mapped in Figure 3?
We normalized the results of L in figure 3 for consistency. We added a sentence to make this more clear.
L183 "over- or underestimation of estimates" -> "over- or underestimation of assessments"
This has been corrected.
L235 "In Eq.3": eq. 4?
This has been corrected.
L257 Eq. 6: the notation is quite confusing. With reference to Eq. 15 in [44] (page 838) the denominator should be [s^b A^c]_m
This has been corrected.
L269 "at seven fixed depth intervals ranging between 0 to 200cm depth." -> "at seven fixed depth intervals ranging between 0 to 200cm."
The word ‘depth’ has been removed.
L276 "Areas where the SFC was low are more susceptible to pesticide fate." -> "Areas where the SFC is low are more susceptible to pesticide fate."
This has been corrected.
L329 Figure 2, caption "pp’DDD stands...": it's not in the figure?
There was a mistake in the figure which has been corrected.
L374 Figure 3: are these values of L of eq. 1?
Yes, we made the link between the equation and the figure more clear throughout the section Results.
L397 I do not understand the sentence "Unless both maps were created using nearly the same methodology, differences could have been cause by the input data and the different resolution of the input data.": if the methodology is the same the different output can only be explained by a difference of input (or, to some extent, of resolution).
We reformulated the sentence to make it more clear.
L461 "Some of these methods were validated, but for most methods a sensitivity analysis of the variables." has been carried out?
Thank you for noticing. The sentence has been completed.
Reviewer 2 Report
1) I failed to find original spatial resolution for used models (Table 1) and it's influence on the modeling results at 5km resolution. If original models have differ (fine) spatial resolution, some upscaling procedure should
be used, especially for features derived from SRTM (slope, flow accumulation, stream length, etc.). We can either calculate features at fine scale, and average them at coarse resolution, or average elevation to coarse resolution, and then calculate features, or fit some recalculation statistical model. Which method was used in the paper? How resolution change influence the results: if we, for example, run model at source resolution, and at 5km resolution, are the results the same? what will be the difference magnitude (in %%)?
2) As I understand, some source models were simplified. How simplification influence the results? what will be the difference magnitude (in %%), caused by simplification?
3) So, we need numerical results to justify the applicability of the selected models, since it makes little sense to discuss models that do not reflect reality, even at a qualitative level (for example, Sedimentation model, lines 387-399). By the way, correlation 0.16 (line 397) - Pearson? Possibly better to use rank (Spearman) correlation, it can be bigger. And RMSD=1.7 looks strange for normalized to [0,1] data.
4) Eq.1 looks strange, it seems that (1-GW)*5 will be written instead of (1-(1-GW*5)), etc.
5) Typos in Figure 2: Soil and Sediment repeated instead of insecticide names.
Author Response
Response to reviewer 2
1) I failed to find original spatial resolution for used models (Table 1) and it's influence on the modeling results at 5km resolution. If original models have differ (fine) spatial resolution, some upscaling procedure should be used, especially for features derived from SRTM (slope, flow accumulation, stream length, etc.). We can either calculate features at fine scale, and average them at coarse resolution, or average elevation to coarse resolution, and then calculate features, or fit some recalculation statistical model. Which method was used in the paper? How resolution change influence the results: if we, for example, run model at source resolution, and at 5km resolution, are the results the same? what will be the difference magnitude (in %%)?
The original spatial resolution of the geospatial data is given in supplementary table S3.
We averaged the variables to coarse resolution and then calculated the different features. We added this to the Materials and Methods section.
2) As I understand, some source models were simplified. How simplification influence the results? what will be the difference magnitude (in %%), caused by simplification?
We did not simplify models. Some processes were described by the simplest model available (linear), because of absence of more complex models or lack in evidence on the relationships between the variables and the mapped process.
3) So, we need numerical results to justify the applicability of the selected models, since it makes little sense to discuss models that do not reflect reality, even at a qualitative level (for example, Sedimentation model, lines 387-399). By the way, correlation 0.16 (line 397) - Pearson? Possibly better to use rank (Spearman) correlation, it can be bigger. And RMSD=1.7 looks strange for normalized to [0,1] data.
If possible, the models were based on existing methods. Literature was used to substantiate the linear models that lacked evidence. We also normalized model output, because we cannot justify a quantitative value to the processes associated with pesticide fate. We tested Spearman correlation, but the correlation did not increase. We corrected the RMSD.
4) Eq.1 looks strange, it seems that (1-GW)*5 will be written instead of (1-(1-GW*5)), etc.
You are absolutely correct. We corrected this.
5) Typos in Figure 2: Soil and Sediment repeated instead of insecticide names.
We corrected the figure.
This manuscript is a resubmission of an earlier submission. The following is a list of the peer review reports and author responses from that submission.
Round 1
Reviewer 1 Report
The authors studied the application of agricultural pesticides and mapped their spatial processes related to pesticide fate in Africa. They want to build a relationship between pesticide fate and key processes. The subject is interesting and relevant to the scope of this journal. However, a main weakness is lack of such a relationship to achieve the objectives. This reviewer recommends to do major revision taking into account all of the below modifications and comments.
1. In Introduction, it is unclear why key geospatial processes affecting environmental fate need to be selected. Are there any differences from those in the fate models? These key processes selected by the authors are well-known. It requires to justify the objectives of this study.
2. In Section 2, it is unclear what relationship is between Section 2.1 and Section 2.3. The authors listed the fate models but did not use them. Then the authors proposed some key fate processes in Section 2.3.
3. In Section 2.2, it should be better to add maps of soil distribution, land cover and slope distribution since they are key input data for leaching and runoff.
4. In Section 2.3, it is unclear how these key processes are related to pesticide fate?
5. Some abbreviations should spend out, such as USLE.
6. In Section 2, it should be better to add a relationship between pesticide fate and these key processes with weight factors.
7. Section 5, the reviewer has not seen any relationship between pesticide fate and these key processes. This is a main weakness.
Reviewer 2 Report
The paper used existing geospatial databases to select major geospatial processes affecting the environmental fate of agricultural pesticides. However, there is a major concern is the modeling process of this paper seems not to follow the standard modeling procedures. The results are not reliable. In other words, there are no procedures of model calibration, parameterization, and validation. Even there is no sensitivity analysis for checking model performance under different settings. The paper used simple linear combinations and ignored the complex physical and non-linear process. Because the authors did not provide any model comparisons or validation results, therefore, the results of the mapping are not convincing and reliable. Authors need to provide more solid evidence or references for supporting their modeling results.
Reviewer 3 Report
L57-65 Can you better clarify the relationship between the research described in this paper and the future development, calibration and validation of pesticide fate models in the African Continent?
L156 is any reference for eq. 1 available in literature or, alternatively, can you elaborate on its form and the parameters used in this equation?
L154 "three processes; the susceptibility": is the semicolon actually a colon? The same situation happens on L219.
L198 why does the equation number restart from 1 (equation at L156 is already eq. 1)?
L224 The threshold to switch between the two equations 4a and 4b is not 0.09 degrees but 9%, i.e. 5.14 degrees. Panagos et al. ([39] in your reference list) is ambiguous in this regard (top of page 119), but its source
Renard, K.G.; Foster, G.R.; Weesies, G.A.; McCool, D.K.; Yoder, D.C. Predicting Soil Erosion by Water: A Guide to Conservation Planning with the Revised Universal Soil Loss Equation (RUSLE); Agricultural Handbook 703; U.S. Government Printing Office: Washington, DC, USA, 1997.
is clear (see page 33).
Also, the equal sign in eq. 4a and 4b should be swapped: when slope is 0.09 the equation to be used is 4a, not 4b. See eq. 1a and 1b in (Panagos et al.) ([39] in your references list) and (Renard et al.), again on page 33.
It is possible that making these two modifications will not make much difference at the scale you are working.
L337 "Sedimentation takes place at locations where water can accumulate.": a high value in the accumulation map indicates that the flow from a large number of SRTM-DEM cells passes through that cell, but the relationship with sedimentation is tenuous because there is no indication of the factors that influence the sedimentation process, mainly the flow speed. An easy enhancement would be to combine the accumulation map with the slope.
L251 The same remark made for eq. 1 at L 156 applies to equation 5.
L268 The same remark made for eq. 1 at L 156 applies to equation 6.
L387 "The standard deviation": of the monthly values during the year?